# Photoinduced copper-catalyzed C–N coupling with trifluoromethylated arenes

Jun Huang[1,3], Qi Gao[1,3], Tao Zhong[1], Shuai Chen[1], Wei Lin[1,2], Jie Han[1] ✉ & Jin Xie [1] ✉

Selective defluorinative functionalization of trifluoromethyl group (–CF$_3$) is an attractive synthetic route to the pharmaceutically privileged fluorine-containing moiety. Herein, we report a strategy based on photoexcited copper catalysis to activate the C–F bond of di- or trifluoromethylated arenes for divergent radical C–N coupling with carbazoles and aromatic amines. The use of different ligands can tune the reaction products diversity. A range of substituted, structurally diverse α,α-difluoromethylamines can be obtained from trifluoromethylated arenes via defluorinative C-N coupling with carbazoles, while an interesting double defluorinative C-N coupling is ready for difluoromethylated arenes. Based on this success, a carbazole-centered PNP ligand is designed to be an optimal ligand, enabling a copper-catalyzed C–N coupling for the construction of imidoyl fluorides from aromatic amines through double C-F bond functionalization. Interestingly, a 1,2-difluoroalkylamination strategy of styrenes is also developed, delivering γ,γ-difluoroalkylamines, a bioisostere to β-aminoketones, in synthetically useful yields. The DFT studies reveal an inner-sphere electron transfer mechanism for Cu-catalyzed selective activation of C(sp$^3$)–F bonds.

The unique physical and biological properties of fluorinated compounds have led to their wide application in pharmaceuticals[1], agrochemicals[2] and materials[3]. With their high lipophilicity, metabolic stability and unique electronic features, CF$_2$ groups have attracted increasing attention (Fig. 1a)[4]. In recent decades, deoxyfluorination[5,6], site-selective fluorination[7–9] and fluoroalkylation[10–15] reactions have been used to construct this interesting motif. Meanwhile, selective functionalization of one C–F bond in trifluoromethyl groups has also gained great momentum because it can provide elegant access to a series of privileged compounds derived from commercially available trifluoromethylated arenes[16–33]. Recently, radical defluoroalkylation[34–36], defluoroarylation[37], defluorohydrogenation[38–40] and defluorocarboxylation[32] reactions have been achieved, all of which constructed C–C or C–H bonds from C–F bonds in a trifluoromethyl group. A very recent work from Xu's group disclosed a radical coupling pathway for the formation of C–X (X = O, S,

Se) bonds with reactive Ar-XH (pK$_a$ -6)[41]. However, to the best of our knowledge, catalytic defluorinative C–N coupling from C–F bond in trifluoromethylated arenes has been reported very rarely (Fig. 1b)[25,42–44].

Remarkably, α,α-difluoromethylamines are bioisosteres for amides[45], and this would be significant in new drug discovery. Several issues, however, have hindered the construction of such structures and these include: (1) the bond dissociation energy (BDE) of C–F bond in CF$_3$ is strong[46] but the BDEs of the remaining C–F bonds significantly decreases once the F atoms have been substituted. For example, the BDEs of the C–F bonds in PhCF$_3$, PhCHF$_2$, and PhCH$_2$F are 118, 107 and 99 kcal mol$^{-1}$, respectively[47], which often contributes to undesired over-defluorination; (2) the reactions of the difluoromethyl radical are generally limited to radical addition or a HAT process and thus construction of CF$_2$–X bond calls for merging strategies such as the combination of transition metal catalysis. For example, Zhang and co-

[1]State Key Laboratory of Coordination Chemistry, Jiangsu Key Laboratory of Advanced Organic Materials, Chemistry and Biomedicine Innovation Center (ChemBIC), School of Chemistry and Chemical Engineering, Nanjing University, Nanjing 210023, China. [2]School of Chemistry and Environmental Engineering, Jiangsu University of Technology, Changzhou 213001, China. [3]These authors contributed equally: Jun Huang, Qi Gao. ✉e-mail: jie.han@nju.edu.cn; xie@nju.edu.cn

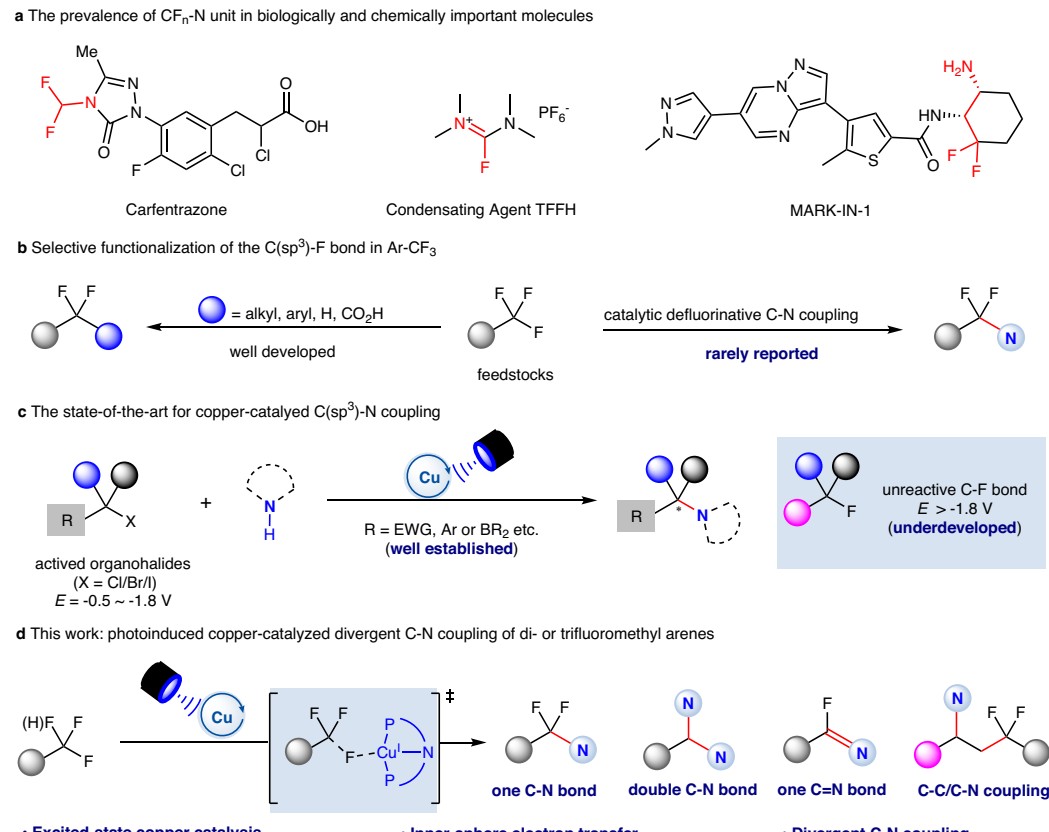

**Fig. 1 | Challenges and strategies in selective defluorinative C−N coupling of the C(sp³)−F bond in Ar-CF₃ and Ar-CHF₂.** a The prevalence of CFₙ-N unit in biologically and chemically important molecules. b Selective functionalization of the C(sp³)-F bond in Ar-CF₃. c The state-of-the-art for copper-catalyzed C(sp³)-N coupling. d This work: photoinduced copper-catalyzed divergent C−N coupling of di- or trifluoromethyl arenes.

workers have reported an elegant photoinduced Pd-catalyzed defluorinative arylation, and the combination of transition metal catalysis unlocked new reaction pathways for selective functionalization of C(sp³)−F bond[37]. In recent years, Cu-mediated radical coupling with light excitation has become an emerging field[48,49]. A series of pioneering copper-catalyzed C−N couplings via a radical pathway using Ar−X (X = Cl, Br, I)[50,51] or activated aliphatic halides (R−X, X = Cl, Br, I)[52–59] has been disclosed by Fu and Liu respectively. However, the activation of more inert C−F bonds[56] to construct C−N bonds is still very challenging (Fig. 1c). Inspired by these seminal works, we wondered if photoinduced copper-catalyzed defluorinative C−N coupling of a CF₃ group could afford difluoromethylated products, whose synthesis otherwise would be difficult.

Here, we report copper-catalyzed divergent defluorinative C−N coupling of trifluoromethylated arenes with carbazoles and aromatic amines. A photoexcited copper-complex can accomplish the inner-sphere electron transfer with the C−F bond in Ar-CF₃ to generate the difluoromethyl radical as key intermediates (Fig. 1d). These radicals can interact with the generated R₂N·Cu^II-species for radical C−N coupling. An interesting double defluorinative C−N coupling has been realized for difluoromethylated arenes. When primary aromatic amines are used as the nitrogen source with a carbazole-centered PNP ligand, a continuous defluorination process would occur to afford versatile imidoyl fluorides in synthetically useful yields. In addition, 1,2-difluoroalkylamination of styrenes can be realized to furnish useful β-aminoketones bioisosteres, γ,γ-difluoroalkylamines. This protocol offers a divergent C−N coupling for the synthesis of synthetically interesting α,α-difluoromethylamines, imidoyl fluorides and γ,γ-difluoroalkylamines from electron-deficient trifluoromethylated arenes and aromatic amines. The broad reaction scope, excellent

functional group tolerance and gram-scale ability enable this strategy to be promising for the construction of value-added products.

## Results
### Reaction optimization
To initiate this study, the defluorinative C−N coupling reaction of 1,3-bis(trifluoromethyl)-benzene (**1a**) and carbazole (**2a**) was selected as the model reaction with which to optimize the reaction conditions (Table 1). The standard conditions include the use of CuBr as catalyst and ⁿBu₃P as the ligand with ᵗBuOLi as an inorganic base under LED irradiation (λ_max = 390 nm). This delivered the desired defluorinated product (**3a**) in 72% isolated yield in 1 h (Table 1, entry 1). It was found

**Table 1 | Optimization of reaction conditions.ᵃ**

| Entry | Variation of standard conditions | Yield |
|---|---|---|
| 1 | none | 73% (72%) |
| 2 | ᵗBuOK instead of ᵗBuOLi | trace |
| 3 | MeCN instead of MTBE | N.D. |
| 4 | Et₂O instead of MTBE | 71% |
| 5 | CuCl instead of CuBr | 72% |
| 6 | CuI instead of CuBr | 14% |
| 7 | no ⁿBu₃P | 13% |
| 8 | no CuBr | N.D. |
| 9 | no light | N.D. |

Standard conditions: CuBr (2 mol%), ⁿBu₃P (4.8 mol%), 1,3-bis(trifluoromethyl)-benzene **1a** (0.5 mmol), carbazole **2a** (0.1 mmol), ᵗBuOLi (0.2 mmol), MTBE (4 mL), LEDs (λ_max = 390 nm), rt, 1 h. GC yield with dodecane as internal standard is shown and isolated yield is shown in parenthesis.
*MTBE* methyl tert-butyl ether, *N.D.* not detected.

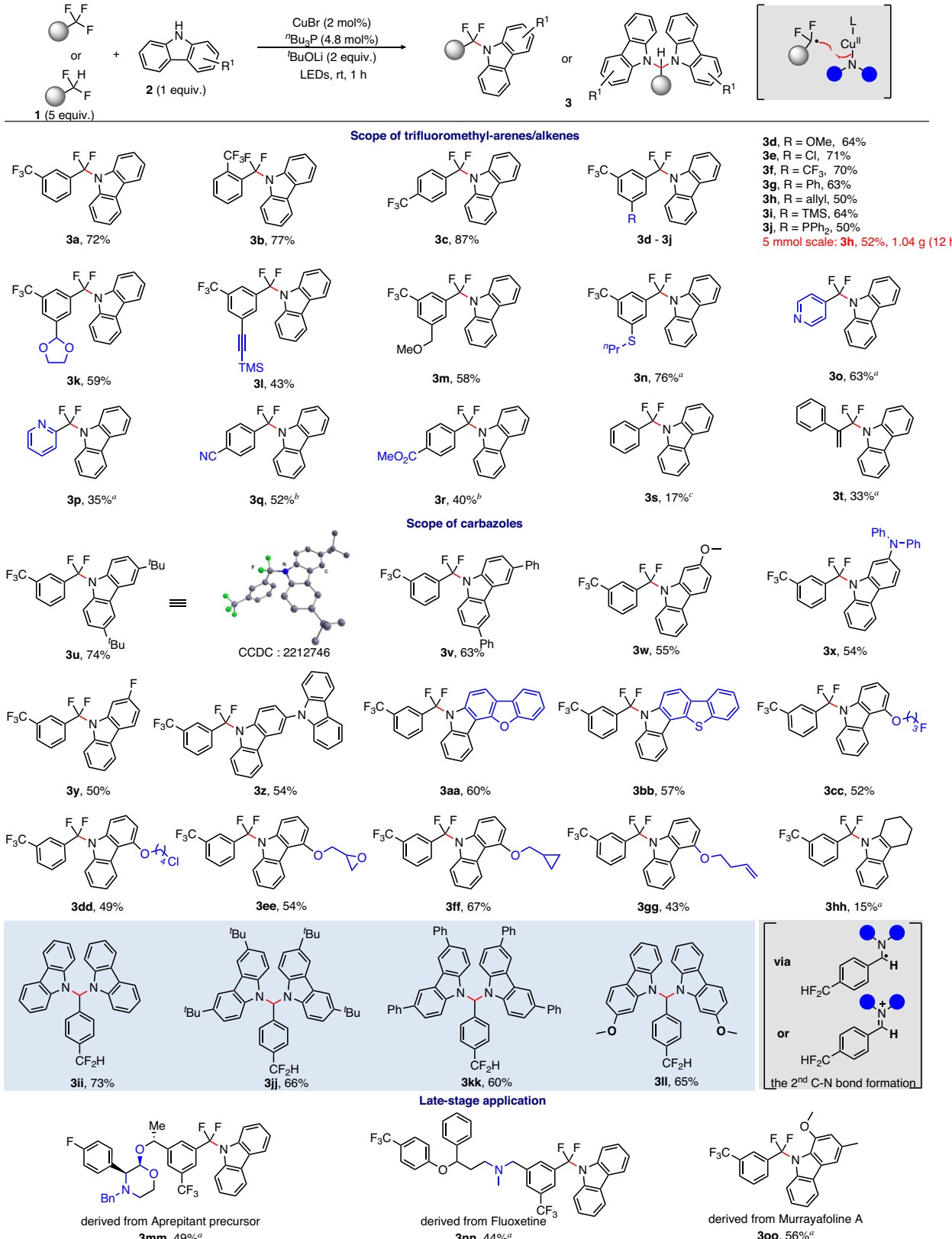

**Fig. 2 | Substrate scope of defluorinative C–N coupling under standard reaction conditions.** [a]CuBr (10 mol%), [n]Bu₃P (24 mol%), 12 h. [b]CuBr (10 mol%), tris(2,4-di-tert-butylphenyl)phosphite (24 mol%), 12 h. [c]CuBr (10 mol%), tris(2,4-di-tert-butylphenyl)phosphite (24 mol%), trifluoromethylbenzene (1 mmol), 12 h. Isolated yields unless otherwise indicated.

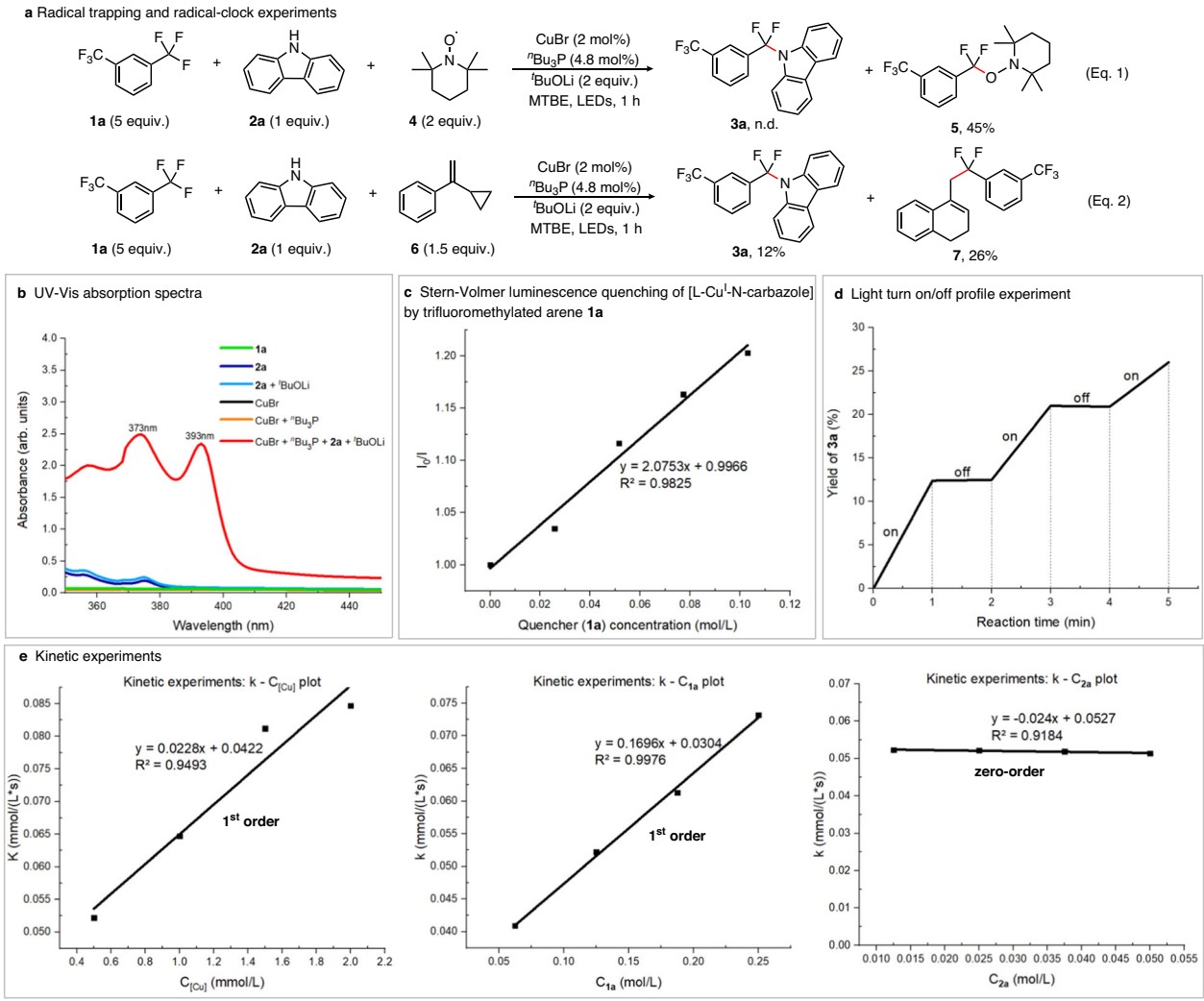

**Fig. 3 | Study of the mechanism for defluorinative C–N coupling with carbazoles. a** Radical trapping and radical-clock experiments. **b** UV-Vis absorption spectra. **c** Stern-Volmer luminescence quenching of [L-Cu$^I$-$N$-carbazole] by trifluoromethylated arene **1a**. **d** Light turn-on/off profile experiment. **e** Kinetic experiments.

that the replacement of $^t$BuOLi with $^t$BuOK resulted in only a trace amount of product (entry 2). It was considered that $^t$BuOLi may have better solubility in methyl *tert*-butyl ether (MTBE) and also that the lithium ion could be a hard acid to facilitate the elimination of the fluoride[60]. A series of different solvents were examined and it was found that ethers, such as diethyl ether and methyl *tert*-butyl ether gave better yields (entries 3 and 4). When CuCl replaced CuBr (entry 5), a comparable yield of 72% was obtained but CuI as catalyst furnished the product (**3a**) in only 14% yield (entry 6). In the absence of $^n$Bu$_3$P as ligand, a decreased yield of 13% was obtained (entry 7). The control experiments showed that this reaction cannot occur in the absence of either the catalyst, CuBr or light irradiation (entries 8 and 9).

## Substrate scope

With the standard reaction conditions in hand, we investigated the scope of substrates. As shown in Fig. 2, a variety of trifluoromethyl arenes can afford the desired C–N coupling products (**3a**–**3s**) in moderate to good yields. A series of synthetically useful functional groups, such as chloride (**3e**), terminal alkene (**3h**), silane (**3i**), alkyne (**3l**), thioether (**3n**), nitrile (**3q**) and ester (**3r**) tolerate the reaction conditions well. The gram-scale experiment was also carried out without compromising the reaction efficiency and the product (**3h**) was obtained in 52% isolated yield at 5 mmol scale. When the

trifluoromethylated arenes bearing a triarylphosphine moiety were subjected to the standard conditions, the reaction was not affected and the desired product (**3j**) was obtained in 50% yield. Heterocyclic substrates such as trifluoromethyl pyridines can also give corresponding products (**3o, 3p**) in moderate yields albeit with an increased catalyst loading (10 mol%) and longer reaction time (12 h). Interestingly, when the arenes bearing an electron-withdrawing group at the *para*-position relative to the trifluoromethyl group were employed, the alkylphosphine ligand ($^n$Bu$_3$P) failed to promote the desired coupling reaction. We speculated that electron-rich ligands are disfavored to the crucial coupling process, thus hindering the formation of products[37,61]. Importantly, an electron-deficient ligand can accelerate this process. When an electron-poor phosphite ester ligand, P(OAr)$_3$ (Ar = 2,4-$^t$Bu$_2$C$_6$H$_3$) was used in place of $^n$Bu$_3$P, the desired products (**3q, 3r**) were formed in moderate yields. When benzotrifluorides was subjected to this protocol, the product (**3s**) could only be obtained in 17% isolated yield. In addition, α-(trifluoromethyl)styrene is also a suitable substrate to deliver the target product (**3t**) in good selectivity.

Subsequently, the scope of the nitrogen-containing partners was investigated. It was found that the carbazoles bearing various functional groups such as methoxy (**3w**), amine (**3x**), fluoride (**3y**), N-, O-, S-containing heteroaromatics (**3z, 3aa, 3bb**), chloride (**3dd**), epoxide (**3ee**) and alkene (**3gg**) all gave the desired products in moderate to

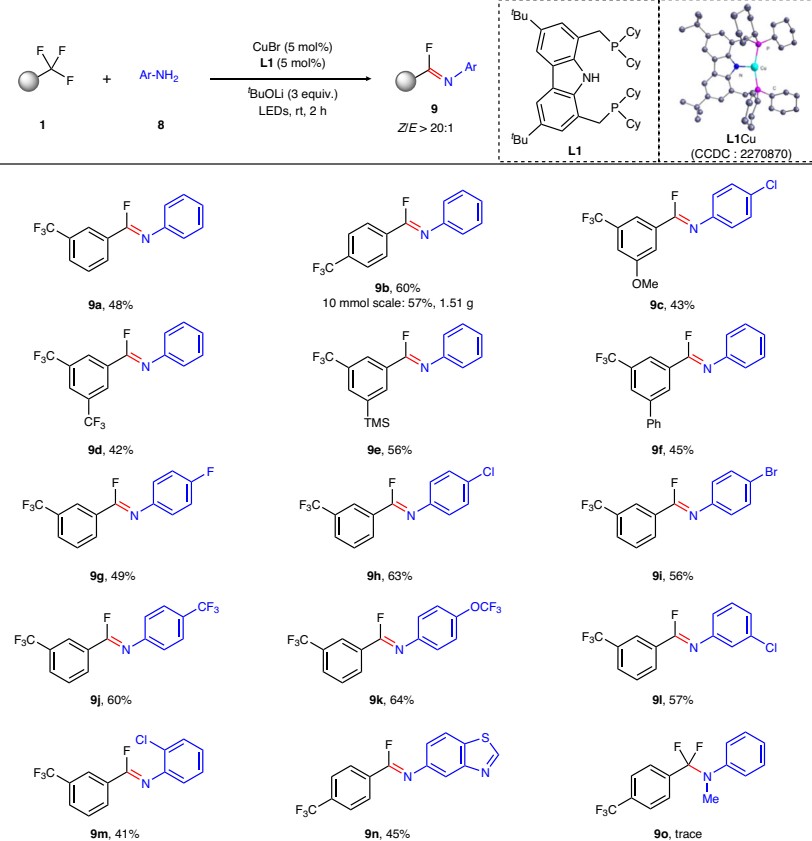

**Fig. 4 | Substrate scope of the tandem C−N coupling and defluorination.** Standard reaction conditions: CuBr (5 mol%), **L1** (5 mol%), trifluoromethylated arenes (0.6 mmol), anilines (0.2 mmol), $^t$BuOLi (0.6 mmol), Et$_2$O (2 mL), LEDs ($\lambda_{max}$ = 390 nm), rt, 2 h. Isolated yields unless otherwise indicated.

good yields. When multisubstituted indole was subjected to the reaction, the product (**3hh**) was produced in only 15% yield. Unfortunately, anilines cannot successfully react in copper-catalyzed defluorinative C−N coupling. We rationalized that the conjugated system in carbazole would facilitate the excitation process of the formed [L-Cu-N-carbazole] species[41], thus promoting the desired defluorinative C−N coupling.

Interestingly, the difluoromethylated arenes are also good coupling partners in Cu-catalyzed defluorinative C−N coupling. Because the difference between the difluoro- and monofluoro-variants, double defluorinative C−N coupling processes could take place for difluoromethylated arenes. Following such reactions, the desired products (**3ii**–**3ll**) were obtained in yields of 60–73%. We speculated the second C−N formation could experience either the radical pathway via the α-*N*-benzylic radical or the ionic mechanism via the iminium ions. Importantly, this protocol can be applied to the late-stage modification of complex trifluoromethylated arenes and carbazoles. As shown in Fig. 2, the Aprepitant precursor, the Fluoxetine derivative and the carbazole alkaloid, Murrayafoline A are good substrates, which can afford the corresponding products (**3mm**–**3oo**) in synthetically useful yields.

## Mechanistic studies

To gain further insight into the reaction mechanism, control experiments were conducted and the results are shown in Fig. 3. To confirm whether a radical process is involved or not, 2,2,6,6-Tetramethylpiperidinooxy (TEMPO) was added into the model reaction (Fig. 3a). As expected, the reaction was completely inhibited and TEMPO-trapped difluorobenzylic radical adduct (**5**) was obtained with 45% isolated yield (Eq. 1). When 1.5 equiv. of (1-cyclopropylvinyl)benzene (**6**) was added to the model reaction under standard conditions, it was found that the desired product (**3a**) was formed in 12% yield and

the radical ring-opening product (**7**) was detected in 26% NMR yield (Eq. 2). Thus, it demonstrates that the reaction proceeds with a radical pathway. The wavelength of experimental LEDs ranged from 360 to 420 nm with $\lambda_{max}$ = ~390 nm (Supplementary Fig. 4). To identify the light-absorbing species in the catalytic cycle, UV-Vis absorption experiments were conducted. It indicated that trifluoromethylated arenes (**1a**), carbazole (**2a**), CuBr, CuBr/$^n$Bu$_3$P or **2a**/$^t$BuOLi all have very weak absorptions in the range of 360–420 nm, while the mixture of CuBr, $^n$Bu$_3$P, carbazole (**2a**) and $^t$BuOLi has significant UV-Vis absorption with two new peaks at 373 and 393 nm (Fig. 3b). Consequently, we speculated that the [L-Cu-N-carbazole] species generated in-situ would absorb the light, and then promote the single electron reduction of trifluoromethylated arenes. Luminescence quenching experiments further confirmed that photoexcited [L-Cu-N-carbazole]* could be quenched by 1,3-bis(trifluoromethyl)-benzene (**1a**) (Fig. 3c). Light turn-on/off control experiments suggested that the reaction requires continuous irradiation of light, and almost no conversion occurred in the absence of light (Fig. 3d). On the other hand, the plots of kinetic experiments suggested a first-order kinetic dependence on [Cu] and [**1a**] but a zero-order kinetic dependence on [**2a**] (Fig. 3e). These mechanistic results indicate that carbazole unit would play a crucial role in this single electron transfer process between photoexcited [L-Cu$^I$-N]* species and trifluoromethylated arenes. Inspired by this, we wondered if we could investigate one phosphine-based ligand bearing one carbazole moiety, which may further extend the N-sources in such defluorinative transformations.

In the light of recent achievements of PNP ligand[53,62–65], we assumed that this skeleton combined with carbazoles and trialkyl phosphines would facilitate the desired defluorinative C−N coupling reaction and further expand the scope of amine substrates. Consequently, carbazole-centered PNP ligand (**L1**) was introduced to achieve

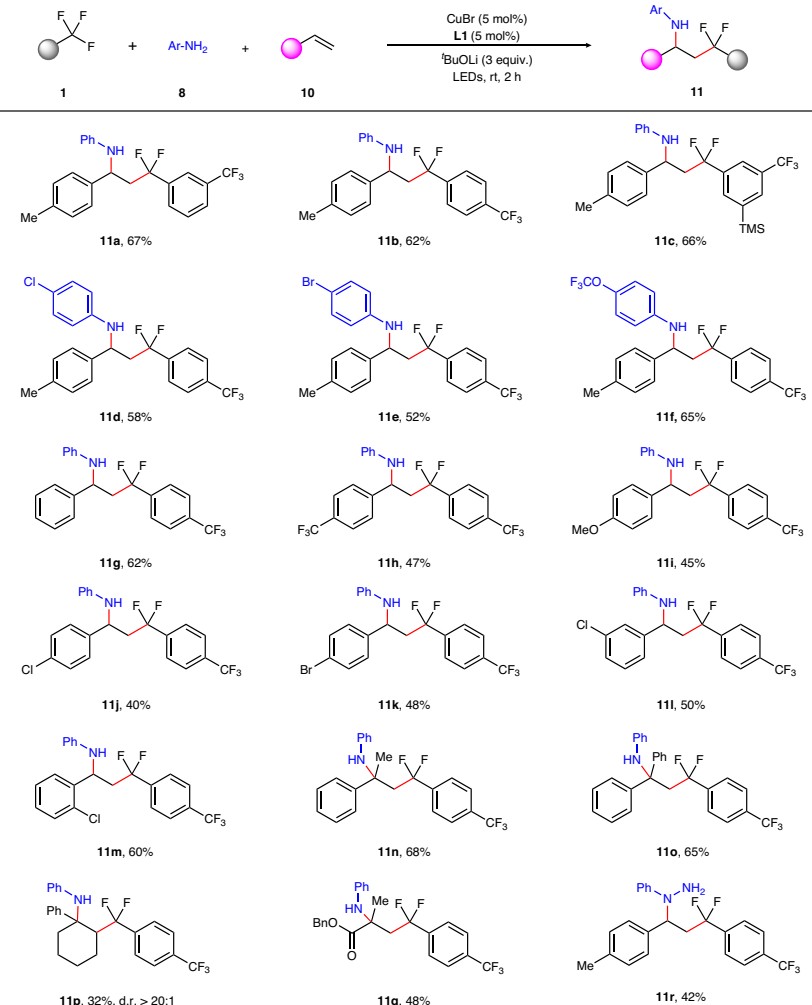

**Fig. 5 | Substrate scope of the 1,2-difluoroalkylamination reaction.** Standard reaction conditions: CuBr (5 mol%), **L1** (5 mol%), trifluoromethyl arenes (0.6 mmol), anilines (0.2 mmol), alkenes (0.6 mmol), $^t$BuOLi (0.6 mmol), Et$_2$O (2 mL), LEDs ($\lambda_{max}$ = 390 nm), rt, 2 h. Isolated yields unless otherwise indicated.

copper-catalyzed C–N coupling of trifluoromethylated arenes and anilines. To our surprise, the resulting α,α-difluoromethylamines was immediately transformed to imidoyl fluorides through tandem C–N coupling and defluorination under basic conditions (Supplementary Table 10)[66]. Despite of the promising synthetic potential of imidoyl fluorides, their synthesis is challenging and in general harsh reaction conditions are required while with limited scope[67–70]. A series of structurally diverse imidoyl fluorides (**9a**–**9n**) can be readily obtained in synthetically useful yields from commercially available trifluoromethylated arenes and aromatic amines as shown in Fig. 4. A 10 mmol scale experiment was also tested and imidoyl fluorides (**9b**) can be obtained in 57% isolated yield, further demonstrating its robust synthetic practicality. It is worth noting that aniline with halogen such as bromide (**9i**) was also compatible with this reaction. The heteroaromatic amines (**9n**) also tolerated this reaction conditions well. When secondary amines such as *N*-methylaniline was employed as the N-based nucleophile, the desired defluorinative C–N coupling product (**9o**) could be detected with ~5% yield by GC-MS.

Provided by the unique reactivity of α,α-difluorobenzylic radical intermediate in the defluorinative C–N coupling process, we questioned if alkenes could be introduced into the reaction system to achieve 1,2-difluoroalkylamination reaction of the alkenes. Once successful, it can afford a new route for the synthesis of γ,γ-difluoroalkylamines, one of the bioisosteres to β-aminoketones. With this consideration, several alkenes were investigated and styrenes were

found to be the most suitable radical receptor (Supplementary Table 17). For the aliphatic alkenes, only trace amount of 1,2-difunctionalization products can be detected and the formation of imidoyl fluorides (**9**) remains the main reaction process. We rationalized that the lower stability of corresponding alkyl radical, which may increase the energy barrier to interact with copper-species[71]. A wide range of useful functional groups such as silane (**11c**), chloride (**11d** and **11j**), bromide (**11e** and **11k**), methoxy (**11i**) were found to be tolerated well, producing expected 1,2-difunctionalized products in moderate yields as shown in Fig. 5. *Ortho*-substituted styrenes (**11 m**) had little influence on the yield. Besides, 1,1-disubstituted olefins (**11n** and **11o**), internal alkenes and electron-withdrawing alkene were also suitable substrates to furnish products (**11p** and **11q**) in moderate yields. In addition, it is interesting to find that phenylhydrazines can proceed this 1,2-difluoroalkylamination to deliver the expected product (**11r**) in 42% yield.

## Synthetic application

To further explore the synthetic utility of this reaction, Ni-catalyzed remote migratory hydroarylation and epoxidation with 3-chloroperoxybenzoic acid (*m*-CPBA) of **3h** gave the target products (**12**, **13**) in 68 and 88% yields, respectively (Fig. 6a). As to imidoyl fluoride **9b**, the corresponding products (**14**, **15**) can be obtained in 91 and 89% yields through the substitution of methoxide and the cyclization reaction of tetrabutylammonium azides. The defluorinative C–N coupling strategy was used to introduce a difluoromethyl unit

**Fig. 6 | Synthetic applications. a** Downstream transformation. **b** Synthesis of Carvedilol derivative.

into the antihypertensive drug carvedilol. As shown in Fig. 6b, product (**16**) can be obtained by stepwise defluorinative C−N coupling and subsequent nucleophilic ring-opening reaction of 2-(2-methoxyphenoxy)ethan-1-amine to the epoxide (**3ee**).

## Theoretical study

A possible mechanism is proposed in Fig. 7a. Upon light irradiation, the catalyst [**L1Cu$^I$**] (**17**) is excited to [**L1Cu$^I$**]* species (**18**), which can undergo either an outer-sphere single electron transfer (OSET) process with the highly electron-deficient trifluoromethylated arene (**1**) to furnish the [**L1Cu$^{II}$**]$^+$ intermediate (**19**) and a radical anion (**20**) or an inner-sphere single electron transfer (ISET) to furnish [**L1Cu$^{II}$-F**] (**21**) and the difluorobenzylic radical (**22**) directly. The radical anion (**20**) from the OSET process would lose one fluoride to produce difluorobenzylic radical (**22**). When styrenes are employed, **22** is easily converted to the more stable secondary benzyl radical (**23**). Through the ligand substitution process, [**L1Cu$^{II}$-F**] (**21**) can be transformed to [**L1Cu$^{II}$-NHAr**] species (**24**), which could interact with the difluorobenzylic radical (**22**) or the benzyl radical (**23**) to give the coupling products **3** or **11**, respectively. Although the difluorobenzylic radical (**22**) or the benzyl radical (**23**) is less likely to proceed radical oxidative addition to generate Cu(III)-intermediate for subsequent reductive elimination[55,56], its possibility to generate coupling products (**3** or **11**) cannot be completely ruled out[72]. Specially, NH in the α,α-difluoromethylamines (**3**) becomes much more acidic with the influence of a difluoromethyl group, and thus the easy elimination of one fluoride would give rise to imidoyl fluoride (**9**) as the final product.

Based on our recent theoretical understanding of the ISET process[73,74] DFT calculations were performed to elucidate the crucial SET process in the proposed mechanism. All computational calculations were carried out by Gaussian 09B (see computational details in Supplementary Information Section 8). As depicted in Fig. 7b, ISET can facilitate one-step generation of [**L1Cu$^{II}$-F**] (**21**) and α,α-difluorobenzylic radical (**22**), featuring a low energy barrier of only 5.8 kcal•mol$^{-1}$ (**TS1**), accompanied with a free energy drop of -13.2 kcal•mol$^{-1}$. Notably, in the precursor for ISET (**INT1**) at triplet state, π-π stacking effect between the PNP ligand and the electron-deficient trifluoromethylated arenes (**1**) facilitates the substrate-catalyst binding

between **L1Cu** and **1**, thus promoting the crucial electron transfer process (Supplementary Fig. 16).

On the other hand, the OSET pathway involved stepwise electron transfer and the dissociation process of the C−F bond. The free energy barrier for the electron transfer was estimated to be 10.7 kcal•mol$^{-1}$, which is higher than that with ISET process. Consequently, the ISET pathway exhibited superior kinetic favorability over the OSET one. It would enrich copper-catalyzed electron transfer mechanism. Further computational analyses were also performed to explore potential pathways of C−N cross-coupling (Supplementary Fig. 16). These calculations indicated that the reaction favored the ISET pathway, followed by MECP (Minimum Energy Crossing-Point)[75] mediated radical capturing (RC) process.

## Discussion

In summary, we have developed a divergent defluorinative C−N coupling reaction of electron-deficient di- and trifluoromethylated arenes and aromatic amines via photoexcited copper catalysis. Highly selective C−N coupling, tandem C−N coupling/defluorination and 1,2-difluoroalkylamination of styrenes have been readily realized under mild reaction conditions. This protocol can afford a variety of structurally diverse α,α-difluoromethylamines, imidoyl fluoride and γ,γ-difluoroalkylamines with good functional group tolerance. DTF studies of the mechanism identify the difluorobenzylic radicals produced via inner-sphere electron transfer as the key intermediates for the subsequent C−N coupling process. This protocol is a progressive step in copper-catalyzed inert C−F bond activation. It should also represent an important step forward to divergent transformations of trifluoromethylated compounds into privileged fluorine-containing moiety.

## Methods

### General procedure for defluorinative C−N coupling of carbazoles

An oven-dried vial (8 mL) equipped with a magnetic stir bar, is charged with CuBr (0.29 mg, 0.002 mmol, 2 mol%), carbazole **2** (0.1 mmol, 1 equiv.), $^t$BuOLi (16.0 mg, 0.2 mmol, 2 equiv.), MTBE (4 mL) and $^n$Bu$_3$P (1.2 μL, 0.0048 mmol, 4.8 mol%). The resulting reaction mixture is

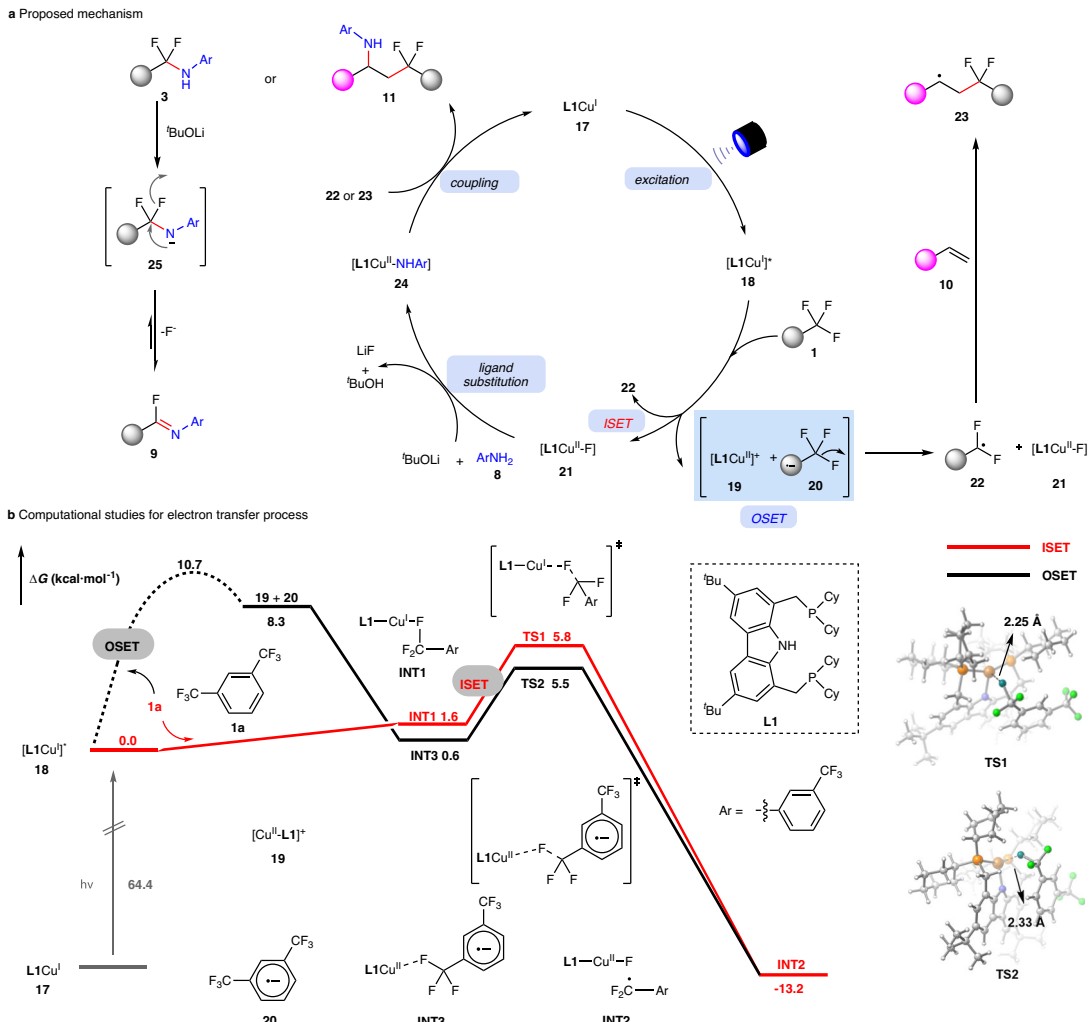

**Fig. 7 | Mechanism proposal. a** Proposed mechanism **b** DFT computed energy profiles for the SET processes between **L1**Cu$^I$ and **1a** at the PBE0(D3BJ)/def2-SVP/def2-TZVP/IEFPCM(ether)//PBE0(D3BJ)/def2-TZVP/SMD(ether) level theory.

allowed to stir at rt for 30 min. Subsequently, the trifluoromethylated arene **1** (0.5 mmol, 5 equiv.) is added. Then the vial is sealed with a rubber cap and is irradiated under 390 nm LEDs for 1 h at rt with vigorous stirring. When the reaction is completed, the mixture is concentrated in vacuo. The crude product can be purified by flash column chromatography on neutral Al$_2$O$_3$ with petroleum ether, ethyl acetate and triethylamine as eluent to afford the desired coupling products.

### General procedure for the tandem C–N coupling and defluorination

An oven-dried vial (4 mL) equipped with a magnetic stir bar, is charged with CuBr (1.43 mg, 0.01 mmol, 5 mol %), **L1** (7.0 mg, 0.01 mmol, 5 mol %), $^t$BuOLi (48.0 mg, 0.6 mmol, 3 equiv.) and Et$_2$O (2 mL). The resulting reaction mixture is allowed to stir at rt for 30 min. Subsequently, the trifluoromethylated arene **1** (0.6 mmol, 3 equiv.) and aromatic amine **8** (0.2 mmol, 1 equiv.) are added sequentially. Then the vial is sealed with a rubber cap and is irradiated under 390 nm LEDs for 2 h at rt with vigorous stirring. When the reaction is completed, the mixture is concentrated in vacuo. The crude product can be purified by flash column chromatography on neutral Al$_2$O$_3$ with petroleum ether and ethyl acetate as eluent to afford the desired coupling products.

### General procedure for the 1,2-difluoroalkylamination reaction

An oven-dried vial (4 mL) equipped with a magnetic stir bar, is charged with CuBr (1.43 mg, 0.01 mmol, 5 mol %), **L1** (7.0 mg,

0.01 mmol, 5 mol %), $^t$BuOLi (48.0 mg, 0.6 mmol, 3 equiv.) and Et$_2$O (2 mL). The resulting reaction mixture is allowed to stir at rt for 30 min. Subsequently, the trifluoromethylated arene **1** (0.6 mmol, 3 equiv.), styrene **10** (0.6 mmol, 3 equiv.) and aromatic amine **8** (0.2 mmol, 1 equiv.) are added sequentially. Then the vial is sealed with a rubber cap and is irradiated under 390 nm LEDs for 2 h at rt with vigorous stirring. When the reaction is completed, the mixture is concentrated in vacuo. The crude product can be purified by flash column chromatography on silica gel with petroleum ether and ethyl acetate as eluent to afford the desired coupling products.

### Data availability

Crystallographic data for the structures reported in this Article have been deposited at the Cambridge Crystallographic Data Centre, under deposition numbers CCDC 2212746 (**3u**), CCDC 2270870 (catalyst **L1**Cu), and CCDC 2281146 (**9b**). Copies of the data can be obtained free of charge via https://www.ccdc.cam.ac.uk/structures/. Data related to materials and methods, optimization of conditions, experimental procedures, mechanistic experiments, and spectra are provided in the Supplementary Information. Source data containing the coordinates of the optimized structures are present. All data are available from the corresponding authors upon request. Source data are provided with this paper.

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

## Acknowledgements

We thank National Key Research and Development Program of China (2022YFA1503200 and 2021YFC2101901), the National Natural Science Foundation of China (22122103, 22101130, and 21971108), Fundamental Research Funds for the Central Universities (020514380304, 020514380252 and 020514380272) for financial support. All theoretical calculations were performed at the High-Performance Computing Center (HPCC) of Nanjing University. N.L., Y.L., and T.Z. at Nanjing University are gratefully acknowledged for their reproduction of the experimental procedures for products **3a**, **3o**, and **11n**.

## Author contributions

J.X. and Jie Han conceived and designed the project. Jun Huang, T.Z., and W.L. performed and analyzed the experimental data. Q.G. performed the density functional theory calculations and discussed the results with S.C. and Jie Han, and J.X. co-wrote the manuscript with input from all the other authors.

## Competing interests

The authors declare no competing interests.
