## [Peer Review File · Nature Communications]

Photoinduced Copper-Catalyzed C–N Coupling with Trifluoromethylated ArenesREVIEWER COMMENTS

Reviewer #1 (Remarks to the Author):

Comments are attached.

Reviewer #2 (Remarks to the Author):

The authors report a photopromoted Cu-catalyzed defluoroamination coupling protocol for electron-deficient trifluoromethylarenes. Carbazoles react using simple Cu/phosphine conditions to provide difluorobenzyl amine products in good yields. Mechanistic studies support a SET pathway from the LCu-carbazole intermediate to generate a difluorobenzyl radical intermediate for amination. The authors used this insight to develop a new ligand system that allowed anilines to do double C–F substitution en route to imidoyl fluorides and for a cascade addition sequence with styrenes. Computational studies support an inner sphere ET pathway for this process.

I am supportive of publication in Nat. Commun. The work expands the scope of monoselective C–F cross-coupling reactions to N-centered nucleophiles. Previous work, published in ACS Catalysis by Xu (ref 35), achieved a similar transformation for phenols and thiols. This present work appears to be a more effective system for anilines and carbazoles, and it also was expanded to a very useful cascade process with styrenes to make beta-amino difluorobenzyl compounds. The scope is generally good, although the trifluoromethylarene must be electron deficient. I think this is also important because it shows how metal complexes can be used to promote these very challenging C–F substitution reactions, and it could lead to other photopromoted C–F “cross coupling” methods (whereas most early work on ArCF₃ C–F functionalization relied on radical “trapping”). I would like the authors to address the points made below before publication.

- Line 47 – explain what you mean by saying “unreactive”
- Line 76 – change the word “remarkably”
- Line 82 – in the intro, please note that this chemistry is limited to electron-deficient ArCF₃, otherwise I find that this intro paragraph is mostly hype and does not serve to accurately describe the work that is going to be discussed
 - o This should be mentioned in line 127 as well
- Line 141 – this description isn’t quite right, the key would be the difference between the difluoro and monofluoro variants, since that’s where the selectivity difference originates
- For the difluoromethylarene scope, please note why only one substrate (bis(difluoromethyl)benzene) is shown
 - o It also seems possible that the double amination process here could be from an ionic mechanism, just forming an iminium ion and having another carbazole add in?
- Figure 4 – please draw out L1, it isn’t obvious what this ligand is

Reviewer #3 (Remarks to the Author):

The report “Photoinduced Copper-Catalyzed Divergent C–N Coupling with Unreactive Trifluoromethyl Groups” primarily details a copper catalyzed photoredox reaction between electron poor benzotrifluorides and carbazole. The work claims to be selective for single C-F

functionalization, but the yields are based on carbazole which is used in five times lower concentration than the fluoride substrates. Despite this, difluoromethyl substrates result in overdefluorination, installing two carbazoles. Further work is performed where i. reaction using only anilines results in C-F substitution followed by H-F elimination to give imidoyl fluorides, and ii. using a combination of styrenes and anilines, an alkene insertion proceeds the reaction with aniline to provide γ -substituted amines. Overall, the work is well performed but very restrictive in its utility given that it is only applicable to very electron deficient benzotrifluorides (used in 5x excess) and carbazole amines. I recommend the work be re-evaluated after major revision or submitted elsewhere (if improvement of the scope isn't possible).

Comments for major revisions

- The reaction is of little use while restricted to carbazoles. The authors should try extending this work to other secondary amines
- Reaction with alkenes/amines (Figure 5) is the most noteworthy work in this report. Can this reaction be extended to other alkenes/amines? E.g. aliphatic amines, Michael acceptor alkenes

Comments for minor revision

- Many recent reviews on selective functionalization of CF₃ groups are missing. Include: Org. Chem. Front. 2021, 8, 3915; Adv. Synth. Catal. 2022, 364, 234; Sci. China Chem. 2021 64, 1630; Org. Biomol. Chem. 2021, 19, 947; Synlett 2020, 31, 933; Chin. J. Org. Chem. 2021, 41, 4554
- The authors state that "to the best of our knowledge, catalytic defluorinative C–N coupling from C–F bond in trifluoromethylated arenes have been reported very rarely (Figure 1b)." But they do not give any references for the rare reports of C-N formation from CF₃ groups. All of Young's reports on selective activation of CF₃ groups involve catalytic formation of C-N bonds (i.e. Angew. Chem. Int. Ed. 2022, 61, e202210917; Chem. Sci. 2023, 14, 1291, J. Am. Chem. Soc. 2020, 142, 2572). Further, a relevant report from Iaroshenko generates C-N bonds from CF₃ non-catalytically, Adv. Synth. Catal. 2021, 363, 54.
- The authors state, "In the light of recent achievements of PNP-ligand,44,53". Reference 44 is an original research paper, and reference 53 is a perspective on T-shaped metalloradicals. There are many better general references for background reading on PNP pincer ligands, for better examples see: Morales-Morales, C. M. Jensen, Elsevier, Amsterdam, The Netherlands, 2007; G. van Koten, J. Organomet. Chem. 2013, 730, 156; E. Peris, R. H. Crabtree, Chem. Soc. Rev. 2018, 47, 1959
- The authors state, "When the trifluoromethylated arenes bearing an electron-donating triarylphosphine [sic] moiety were subjected to the standard conditions, the reaction was not affected" This is wrong. Diaryl phosphino groups are electron withdrawing (see: Chem. Rev. 1991, 91, 165). Besides, it would be misleading to state that the reaction tolerates EDGs when the authors have attempted to install a meta-position PPh₂ (it is actually more EW in the para position due to poor p-p overlap) and when an auxiliary CF₃ is still necessary for the reaction. The reported reaction is restricted to very electron deficient benzotrifluorides.
- The authors state, "When trifluoromethyl benzene was subjected to this protocol, a moderate yield of the product (3s) was obtained." 17% is not a moderate yield. This should be described as poor. Can the authors also specify if yields provided are isolated or GC yields for Figure 2 in the Figure description
- Page 1: emerging strategies  should be 'merging strategies'
- "and the combination of transition metal catalysis unlocked new reaction pathways for selective functionalization of C(sp³)–F bond." Add reference 31 to the end of the sentence
- In Figure 1, TFFA is a fluorination reagent (it wouldn't be stable under biological

conditions). Perhaps the authors could rephrase the description as 'The prevalence of CF_n-N unit in biologically and chemically important molecules'

- In Figure 1, d: Wok  Work

- Write out methyl tertbutyl ether (MTBE) upon first usage. The abbreviation is used before the full name

- Gram-scale experiments was  should be 'were'

- "As expected, the reaction was completely prohibited"  should be 'inhibited'

- Add chem draw scheme to Figure 4 for L1 (the molecular structure is hard to discern)

- "successful"  successful

- "it can afford a new route for the synthesis of γ,γ -difluoroalkylamines, which should be esteemed as the bioisostere to β -aminoketones" Please rewrite this sentence. I have no idea what the authors are trying to say.

In this manuscript, the authors realized the selective C–N coupling of C(sp³)–F bonds in di/trifluoromethylated arenes via photoexcited copper catalysis, providing an efficient method for the construction of high-value-added fluorinated compounds. A wide range of ArCF₃ with diverse functional groups can participate in the defluorinative C–N coupling with various carbazoles and arylamines through the use of different ligands. Moreover, a variety of γ,γ -difluoroalkylamines can be obtained via a 1,2-difluoroalkylation process of styrenes, which further demonstrated the synthetic application of this strategy. The DFT studies revealed an inner-sphere electron transfer process between [L¹Cu^I]^{*} and ArCF₃. The manuscript can be published in Nature Communications after minor revisions.

1. In this work, only aryl substituted trifluoromethyl compounds, which are unactivated substrates in Figure 2, can be tolerated in this strategy. Therefore, the description of “unreactive trifluoromethyl groups” in the title and text should be modified.
2. In 2021, the Yu group has realized the visible-light photoredox-catalyzed radical defluorocarboxylation of C(sp³)–F bonds with CO₂ (*Chem* **2021**, 7, 3099-3113), which should also be presented in the background Introduction and Figure 1b.
3. On page 4, the yield corresponding to CuCl is not consistent with the yield shown in Table 2. Please double check the manuscript carefully.
4. The mass balance in the reactions has not been discussed. What kinds of by-products are formed in the reaction optimization process or scope studies?
5. Although a range of primary arylamines are suitable for the reaction, it will be interesting to know what happened with *N*-methylaniline. Whether the C–N coupling happened with *N*-methylaniline?
6. In Figure 5, for the substrate scope of amines, have the authors tried other nitrogen nucleophiles instead of aromatic amines, such as nitrogen-containing heterocycles or hydrazines?
7. A related review (*Sci. China Chem.* **2021**, 64, 1630-1659) should also be cited.

Point-by-point response to reviewers

Reviewer #1 (Remarks to the Author):

In this manuscript, the authors realized the selective C–N coupling of C(sp³)–F bonds in di/trifluoromethylated arenes via photoexcited copper catalysis, providing an efficient method for the construction of high-value-added fluorinated compounds. A wide range of ArCF₃ with diverse functional groups can participate in the defluorinative C–N coupling with various carbazoles and arylamines through the use of different ligands. Moreover, a variety of γ,γ -difluoroalkylamines can be obtained via a 1,2-difluoroalkylation process of styrenes, which further demonstrated the synthetic application of this strategy. The DFT studies revealed an inner-sphere electron transfer process between [L1Cu^I]* and ArCF₃. **The manuscript can be published in Nature Communications after minor revisions.**

Thank you very much for your positive evaluation to the synthetic values of our new synthetic methodology. Much appreciated!

The following points need to address before the publications

1. In this work, only aryl substituted trifluoromethyl compounds, which are unactivated substrates in Figure 2, can be tolerated in this strategy. Therefore, the description of “unreactive trifluoromethyl groups” in the title and text should be modified.

Answer: Thank you very much for your important suggestions. According to your suggestion, the title has been changed to “Photoinduced Copper-Catalyzed Divergent C–N Coupling with Trifluoromethylated Arenes”; the description in the Introduction has been changed to “This protocol offers a divergent C–N coupling for the synthesis of synthetically interesting α,α -difluoromethylamines, imidoyl fluorides and γ,γ -difluoroalkylamines from electron-deficient trifluoromethylated arenes and aromatic amines”; the description in the Conclusion has been modified as “In summary, we have developed a divergent defluorinative C–N coupling reaction of electron-deficient di- and trifluoromethylated arenes and aromatic amines via photoexcited copper

catalysis”.

2. In 2021, the Yu group has realized the visible-light photoredox-catalyzed radical defluorocarboxylation of C(sp³)-F bonds with CO₂ (*Chem* **2021**, 7, 3099-3113), which should also be presented in the background Introduction and Figure 1b.

Answer: Thanks for your suggestion. We have added the literature (*Chem* **2021**, 7, 3099-3113) in the background Introduction and Figure 1b.

Change in Figure 1b:

b Selective functionalization of the C(sp³)-F bond in Ar-CF₃

Accordingly, the introduction was changed to “Recently, radical defluoroalkylation,³⁴⁻³⁶ defluoroarylation,³⁷ defluorohydrogenation³⁸⁻⁴⁰ and defluorocarboxylation³² reactions have been achieved”.

3. On page 4, the yield corresponding to CuCl is not consistent with the yield shown in Table 2. Please double check the manuscript carefully.

Answer: Thank you for your attention to detail. Upon re-examination, we confirm that the GC yield of **3a** was 72% when CuCl served as the catalyst.

4. The mass balance in the reactions has not been discussed. What kinds of by-products are formed in the reaction optimization process or scope studies?

Answer: Thanks for your valuable suggestion. The major by-product was the dimer of the corresponding difluorobenzyl radical, but its yield typically remained below 5%. According to your suggestion, one new sentence was added in the revised Supplementary Information as “Although the dimer of the corresponding difluorobenzyl radical could be detected as the major by-product, its yield typically remained below 5%”.

As to the 1,2-difluoroalkylation reaction shown in Figure 5, the major by-product was the corresponding imidoyl fluoride and its yield in the reaction optimization process was shown in the revised Supplementary Table 16.

Supplementary Table 16. Screening of the ratio of substrates^[a]

Entry	1a:8a:10a	Yield of 11a ^[b]	Yield of 9a ^[b]
1	3:1:3	71%(67%^[c])	8%
2	3:1:2	60%	11%
3	3:1:1	45%	15%
4	2:1:3	53%	7%
5	1:2:3	30%	5%

[a] Reaction conditions: trifluoromethyl-arenes, anilines, alkenes, **L1Cu** (0.005 mmol, 5 mol%), *t*BuOLi (0.3 mmol, 3.0 equiv.), Et₂O (1.0 mL, 0.1M), 390 nm, rt, 2 h. [b] GC yield with dodecane as internal standard. [c] Isolated yield.

5. Although a range of primary arylamines are suitable for the reaction, it will be interesting to know what happened with *N*-methylaniline. Whether the C–N coupling happened with *N*-methylaniline?

Answer: Thank you very much for your important suggestions. When *N*-methylaniline was employed as the N nucleophile, the desired defluorinative C–N coupling product could be detected with ~ 5% yield by GC-MS. However, despite our efforts, we were unable to further improve its yield.

GC-MS spectra was shown as following:

6. In Figure 5, for the substrate scope of amines, have the authors tried other nitrogen nucleophiles instead of aromatic amines, such as nitrogen-containing heterocycles or hydrazines?

Answer: Thank you very much for your thoughtful suggestions. We have explored the use of nitrogen-containing heterocycles such as pyrroles and imidazoles. Unfortunately, they failed to give the desired products. On a positive note, we found that phenylhydrazine was suitable for the desired 1,2-difluoroalkylation, and we have incorporated the corresponding product into Figure 5 as **11r**.

7. A related review (*Sci. China Chem.* **2021**, *64*, 1630-1659) should also be

cited.

Answer: Thank you very much for your helpful comments. We have added the literature in the revised manuscript.

19. Ai, H.-J., Ma, X., Song, Q. & Wu, X.-F. C-F bond activation under transition-metal-free conditions. *Sci. China Chem.* **64**, 1630-1659 (2021).

Reviewer #2 (Remarks to the Author):

The authors report a photopromoted Cu-catalyzed defluoroamination coupling protocol for electron-deficient trifluoromethylarenes. Carbazoles react using simple Cu/phosphine conditions to provide difluorobenzyl amine products in good yields. Mechanistic studies support a SET pathway from the LCu-carbazole intermediate to generate a difluorobenzyl radical intermediate for amination. The authors used this insight to develop a new ligand system that allowed anilines to do double C-F substitution en route to imidoyl fluorides and for a cascade addition sequence with styrenes. Computational studies support an inner sphere ET pathway for this process.

I am supportive of publication in Nat. Commun. The work expands the scope of monoselective C-F cross-coupling reactions to N-centered nucleophiles. Previous work, published in ACS Catalysis by Xu (ref 35), achieved a similar transformation for phenols and thiols. This present work appears to be a more effective system for anilines and carbazoles, and it also was expanded to a very useful cascade process with styrenes to make beta-amino difluorobenzyl compounds. The scope is generally good, although the trifluoromethylarene must be electron deficient. **I think this is also important because it shows how metal complexes can be used to promote these very challenging C-F substitution reactions, and it could lead to other photopromoted C-F "cross coupling" methods (whereas most early work on ArCF₃ C-F functionalization relied on radical "trapping").**

Thank you very much for your positive evaluation to the synthetic values of our new synthetic methodology. Much appreciated!

I would like the authors to address the points made below before publication.

- Line 47 – explain what you mean by saying “unreactive”

Answer: Since the C(sp³)-F bond in CF₃ was one of the strongest single bonds to carbon (BDE>80 kcal mol⁻¹), it was rather inert under common reaction conditions. According to your suggestion, we have made the corresponding changes as “(1) the bond dissociation energy (BDE) of C-F bond in CF₃ is strong⁴⁶ but the BDE of the remaining C-F bonds significantly decreases once the F atoms have been substituted.”.

- Line 76 – change the word “remarkably”

Answer: Thanks for your advice. We have changed “remarkably” in the revised manuscript to “In addition”.

- Line 82 – in the intro, please note that this chemistry is limited to electron-deficient ArCF₃, otherwise I find that this intro paragraph is mostly hype and does not serve to accurately describe the work that is going to be discussed;

This should be mentioned in line 127 as well

Answer: Thanks for your thoughtful suggestions. According to your suggestion, we have made the following changes as “This protocol offers a divergent C-N coupling for the synthesis of synthetically interesting α,α -difluoromethylamines, imidoyl fluorides and γ,γ -difluoroalkylamines from electron-deficient trifluoromethylated arenes and aromatic amines. The broad reaction scope, excellent functional group tolerance and gram-scale ability enable this strategy promising for the construction of value-added products”in the line 82. In line 127, we have reorganized the sentence as “When benzotrifluorides were subjected to this protocol, the product (3s) could only be obtained in 17% isolated yield”.

- Line 141 – this description isn’t quite right, the key would be the difference

between the difluoro and monofluoro variants, since that's where the selectivity difference originates

Answer: Thank you very much for your helpful suggestions. According to your suggestion, we have adjusted the description as "Because the difference between the difluoro- and monofluoro-variants, double defluorinative C-N coupling processes could take place for difluoromethylated arenes"

- For the difluoromethylarene scope, please note why only one substrate (bis(difluoromethyl)benzene) is shown; It also seems possible that the double amination process here could be from an ionic mechanism, just forming an iminium ion and having another carbazole add in?

Answer: Thanks for your reminder. Considering that there are more commercially available trifluoromethylated arenes than difluoromethylated ones and the compound with remaining fluorine atoms may be of more synthetic use, we didn't try that a lot of difluoromethylarenes. The formation of the corresponding iminium ion is also rational and could not be ruled out. The sentence was rewritten as "We speculated the second C-N formation could experience either the radical pathway via the α -N-benzylic radical or the ionic mechanism via the iminium ions". The possibility of ionic mechanism was added to Figure 2.

Changes in Figure 2:

- Figure 4 – please draw out L1, it isn't obvious what this ligand is

Answer: Thank you very much for your suggestion. We have added the structure of **L1** in Figure 4 of the revised manuscript.

Changes in Figure 4:

Reviewer #3 (Remarks to the Author):

The report "Photoinduced Copper-Catalyzed Divergent C–N Coupling with Unreactive Trifluoromethyl Groups" primarily details a copper catalyzed photoredox reaction between electron poor benzotrifluorides and carbazole. The work claims to be selective for single C-F functionalization, but the yields are based on carbazole which is used in five times lower concentration than the fluoride substrates. Despite this, difluoromethyl substrates result in overdefluorination, installing two carbazoles. Further work is performed where i. reaction using only anilines results in C-F substitution followed by H-F elimination to give imidoyl fluorides, and ii. using a combination of styrenes and anilines, an alkene insertion proceeds the reaction with aniline to provide γ -substituted amines. Overall, the work is well performed but very restrictive in its utility given that it is only applicable to very electron deficient benzotrifluorides (used in 5x excess) and carbazole amines. I recommend the work be re-evaluated after major revision or submitted elsewhere (if improvement of the scope isn't possible).

Thank you very much for your positive evaluation to the synthetic values of our new synthetic methodology. Much appreciated!

Comments for major revisions

- The reaction is of little use while restricted to carbazoles. The authors should try extending this work to other secondary amines

Answer: Thanks for your sincere advice. *N*-methylaniline was chosen as the representative secondary amine, the desired defluorinative C–N coupling product could be detected with ~ 5% yield by GC-MS. Unfortunately, attempt to further improve its yield with the optimization of other bases and solvents

failed. We speculated that the rigid ligand skeleton hindered the approaching of secondary amines to the copper catalyst center, thereby reducing the efficiency of the desired C-N coupling reaction.

Entry	Base	Solvent	GC Yield
1	t BuOLi	Et ₂ O	5%
2	NaH (60% in oil)	Et ₂ O	7%
3	n BuLi (2.5 M in hexane)	Et ₂ O	n.d.
4	LiHMDS (1.0 M in THF)	Et ₂ O	n.d.
5	NaH (60% in oil)	toluene	4%
6	NaH (60% in oil)	benzene	4%

- Reaction with alkenes/amines (Figure 5) is the most noteworthy work in this report. Can this reaction be extended to other alkenes/amines? E.g. aliphatic amines, Michael acceptor alkenes

Answer: Thank you very much for your helpful suggestions. When internal alkenes (1-Phenyl-1-cyclohexene) and Michael acceptor alkenes (benzyl methacrylate) were subjected to this protocol, the desired 1,2-difluoroalkylamination products **11p** and **11q** could be obtained with moderate yields. What's more, phenylhydrazine was also the suitable N nucleophile and was transformed to **11r** in 42% yield. As to aliphatic amines, their strong coordination ability may deactivate the catalyst **L1Cu** and the reactions were therefore inhibited.

Successful examples

11p, 32%, d.r. > 20:1

11q, 48%

11r, 42%

Failed examples of aliphatic amines

N.R.

N.R.

N.R.

N.R.

Comments for minor revision

- Many recent reviews on selective functionalization of CF₃ groups are missing. Include: *Org. Chem. Front.* 2021, 8, 3915; *Adv. Synth. Catal.* 2022, 364, 234; *Sci. China Chem.* 2021 64, 1630; *Org. Biomol. Chem.* 2021, 19, 947; *Synlett* 2020, 31, 933; *Chin. J. Org. Chem.* 2021, 41, 4554

Answer: Thank you very much for your helpful suggestions. According to your suggestion, we have added the references in the manuscript as following:

17. Gupta, R., Jaiswal, A. K., Mandal, D. & Young, R. D. A Frustrated Lewis Pair Solution to a Frustrating Problem: Mono-Selective Functionalization of C–F Bonds in Di- and Trifluoromethyl Groups. *Synlett* **31**, 933-937 (2020).

18. Yan, G., Qiu, K. & Guo, M. Recent advance in the C–F bond functionalization of trifluoromethyl-containing compounds. *Org. Chem. Front.* **8**, 3915-3942 (2021).

19. Ai, H.-J., Ma, X., Song, Q. & Wu, X.-F. C-F bond activation under transition-metal-free conditions. *Sci. China Chem.* **64**, 1630-1659 (2021).

20. Carvalho, D. R. & Christian, A. H. Modern approaches towards the synthesis of geminal difluoroalkyl groups. *Org. Biomol. Chem.* **19**, 947-964 (2021).

21. An, X. et al. Recent Advances in the Single C–F Bond Cleavage Reactions of Trifluoromethylarenes. *Chin. J. Org. Chem.* **41**, 4554-4564 (2021).

22. Zhao, F., Zhou, W. & Zuo, Z. Recent Advances in the Synthesis of Difluorinated Architectures from Trifluoromethyl Groups. *Adv. Synth. Catal.*

364, 234-267 (2022).

- The authors state that "to the best of our knowledge, catalytic defluorinative C–N coupling from C–F bond in trifluoromethylated arenes have been reported very rarely (Figure 1b)." But they do not give any references for the rare reports of C-N formation from CF₃ groups. All of Young's reports on selective activation of CF₃ groups involve catalytic formation of C-N bonds (i.e. *Angew. Chem. Int. Ed.* 2022, 61, e202210917; *Chem. Sci.* 2023, 14, 1291, *J. Am. Chem. Soc.* 2020, 142, 2572). Further, a relevant report from Iaroshenko generates C-N bonds from CF₃ non-catalytically, *Adv. Synth. Catal.* 2021, 363, 54.

Answer: Thanks for your advice. According to your suggestion, we have added the references in the manuscript as following:

25. Mandal, D., Gupta, R., Jaiswal, A. K. & Young, R. D. Frustrated Lewis-Pair-Mediated Selective Single Fluoride Substitution in Trifluoromethyl Groups. *J. Am. Chem. Soc.* **142**, 2572-2578 (2020).

42. Mkrtchyan, S. et al. Mechanochemical Transformation of CF₃ Group: Synthesis of Amides and Schiff Bases. *Adv. Synth. Catal.* **363**, 5448-5460 (2021).

43. Khanapur, S. et al. Fluorine-18 Labeling of Difluoromethyl and Trifluoromethyl Groups via Monoselective C–F Bond Activation. *Angew. Chem. Int. Ed.* **61**, e202210917 (2022).

44. Gupta, R., Csókás, D., Lyea, K. & Young, R. D. Experimental and computational insights into the mechanism of FLP mediated selective C–F bond activation. *Chem. Sci.* **14**, 1291-1300 (2023).

- The authors state, "In the light of recent achievements of PNP-ligand,^{44,53}". References 44 is an original research paper, and reference 53 is a perspective on T-shaped metalloradicals. There are many better general references for background reading on PNP pincer ligands, for better examples see: Morales-Morales, C. M. Jensen, Elsevier, Amsterdam, The Netherlands, 2007; G. van Koten, J. Organomet.

Chem. 2013, 730, 156; E. Peris, R. H. Crabtree, Chem. Soc. Rev. 2018, 47, 1959

Answer: Thanks for your helpful suggestions. According to your suggestion, we have added the references in the manuscript as following:

62. Morales-Morales, D. & Jensen, C. M. (ed.) *The Chemistry of Pincer Compounds* (Elsevier, 2007).

63. Koten, G. V. Pincer ligands as powerful tools for catalysis in organic synthesis. *J. Organomet. Chem.* **730**, 156-164 (2013).

64. Peris, E. & Crabtree, R. H. Key factors in pincer ligand design. *Chem. Soc. Rev.* **47**, 1959-1968 (2018).

- The authors state, "When the trifluoromethylated arenes bearing an electron-donating triarylphosphine [sic] moiety were subjected to the standard conditions, the reaction was not affected" This is wrong. Diaryl phosphino groups are electron withdrawing (see: Chem. Rev. 1991, 91, 165). Besides, it would be misleading to state that the reaction tolerates EDGs when the authors have attempted to install a meta-position PPh₂ (it is actually more EW in the para position due to poor p-p overlap) and when an auxiliary CF₃ is still necessary for the reaction. The reported reaction is restricted to very electron deficient benzotrifluorides.

Answer: Thank you very much for your helpful comments. We have changed the description to "When the trifluoromethylated arenes bearing a triarylphosphine moiety were subjected to the standard conditions".

- The authors state, "When trifluoromethyl benzene was subjected to this protocol, a moderate yield of the product (3s) was obtained." 17% is not a moderate yield. This should be described as poor. Can the authors also specify if yields provided are isolated or GC yields for Figure 2 in the Figure description

Answer: Thanks for your advice. We have modified the sentence as "When benzotrifluorides was subjected to this protocol, the product (**3s**) could only be obtained in 17% isolated yield". The footnote "Isolated yields unless otherwise indicated" has also been added below Figure 2.

- Page 1: emerging strategies  should be 'merging strategies'

Answer: Thanks for your reminder. We have corrected the wrong spelling. The sentence has been changed to "the reactions of the difluoromethyl radical are generally limited to radical addition or a HAT process and thus construction of CF₂-X bond calls for merging strategies such as the combination of transition metal catalysis".

- "and the combination of transition metal catalysis unlocked new reaction pathways for selective functionalization of C(sp³)-F bond." Add reference 31 to the end of the sentence

Answer: Thanks for your advice. We have added the reference 31 to the end of the sentence as following:

"For example, Zhang and co-workers have reported an elegant photoinduced Pd-catalyzed defluorinative arylation, and the combination of transition metal catalysis unlocked new reaction pathways for selective functionalization of C(sp³)-F bond.³⁷"

- In Figure 1, TFFA is a fluorination reagent (it wouldn't be stable under biological conditions). Perhaps the authors could rephrase the description as 'The prevalence of CF_n-N unit in biologically and chemically important molecules'

Answer: Thanks for your advice. We have changed the description as "The prevalence of CF_n-N unit in biologically and chemically important molecules".

Changes in Figure 1a:

Carfentrazone

Condensating Agent TFFA

MARK-IN-1

- In Figure 1, d: Wok  Work

Answer: Thanks for your advice. We have corrected the wrong spelling.

- Write out methyl tertbutyl ether (MTBE) upon first usage. The abbreviation is used before the full name

Answer: Thank you very much for your helpful suggestion. We have added "methyl tertbutyl ether" before its abbreviation.

- Gram-scale experiments was  should be 'were'

Answer: Thanks for your suggestion. We have adjusted the statement as "The gram-scale experiment was also carried out without compromising the reaction efficiency and the product (**3h**) was obtained in 52% isolated yield at 5 mmol scale".

- "As expected, the reaction was completely prohibited"  should be 'inhibited'

Answer: Thank you very much for your suggestion. We have corrected this wrong spelling.

- Add chem draw scheme to Figure 4 for L1 (the molecular structure is hard to discern)

Answer: Thank you very much for your suggestion. We have added the structure of **L1** in Figure 4.

Changes in Figure 4:

- "succesful"  successful

Answer: Thank you very much for your suggestion. We have corrected this

wrong spelling.

- "it can afford a new route for the synthesis of γ,γ -difluoroalkylamines, which should be esteemed as the bioisostere to β -aminoketones" Please rewrite this sentence. I have no idea what the authors are trying to say.

Answer: Thanks for your advice. We have rewritten the sentence as "Once successful, it can afford a new route for the synthesis of γ,γ -difluoroalkylamines, one of the bioisosteres to β -aminoketones"

REVIEWERS' COMMENTS

Reviewer #1 (Remarks to the Author):

The authors have addressed my comments nicely and I suggest acceptance of this work. Congratulations to the authors!

Reviewer #2 (Remarks to the Author):

The authors did a good job responding to my comments and the comments of Reviewer 1.

I do not think the authors were able to address the major concerns of Reviewer 3, which I tend to agree with. The most major concern was that the first C-F coupling reaction is limited to carbazoles, making the products not very useful or desired. Based on how the substrate scope tables are shown, it was not obvious to readers that the stoichiometry of this reaction was 5 equivalents of trifluoromethylarene to 1 equivalent of amine coupling partner; this information is absent in Figure 2, the main substrate scope. The other two transformations require 3 equivalents of the trifluoromethylarene. These issues diminish the impact of the work.

On balance, I am still supportive of publication, mainly due to the three component cascade reported in Figure 5. However, if Reviewer 3 is still not satisfied, then this work should probably be submitted to a more focused synthetic journal due to the major scope limitations.

The authors must note in revision the performance of N-alkyl anilines that were described in response to reviewer 1.

Reviewer #3 (Remarks to the Author):

The authors have made a committed effort to resolve the issues raised by the reviewers. Although other secondary amines could not be realised as substrates for this reaction, the authors succeeded in broadening the reaction scope of the cascade reaction beyond styrene derivatives as well as demonstrating that hydrazines served as suitable N sources. I can recommend publication in its current form.

Point-by-point response to reviewers

Reviewer #1 (Remarks to the Author):

The authors have addressed my comments nicely and I suggest acceptance of this work. Congratulations to the authors!

Thank you very much for your positive comments.

Reviewer #2 (Remarks to the Author):

The authors did a good job responding to my comments and the comments of Reviewer 1.

I do not think the authors were able to address the major concerns of Reviewer 3, which I tend to agree with. The most major concern was that the first C-F coupling reaction is limited to carbazoles, making the products not very useful or desired. Based on how the substrate scope tables are shown, it was not obvious to readers that the stoichiometry of this reaction was 5 equivalents of trifluoromethylarene to 1 equivalent of amine coupling partner; this information is absent in Figure 2, the main substrate scope. The other two transformations require 3 equivalents of the trifluoromethylarene. These issues diminish the impact of the work.

On balance, I am still supportive of publication, mainly due to the three component cascade reported in Figure 5. However, if Reviewer 3 is still not satisfied, then this work should probably be submitted to a more focused synthetic journal due to the major scope limitations.

The authors must note in revision the performance of N-alkyl anilines that were described in response to reviewer 1.

Much appreciated for your positive comments.

Answer: According to your suggestion, the stoichiometry of trifluoromethylated arenes has been added in Fig. 2.

The description of the performance of N-alkyl anilines was modified as "When secondary amines such as *N*-methylaniline was employed as the N-based nucleophile, the desired defluorinative C–N coupling product could be detected with ~5% yield by GC-MS." in the revised manuscript.

Reviewer #3 (Remarks to the Author):

The authors have made a committed effort to resolve the issues raised by the reviewers. Although other secondary amines could not be realized as substrates

for this reaction, the authors succeeded in broadening the reaction scope of the cascade reaction beyond styrene derivatives as well as demonstrating that hydrazines served as suitable N sources. **I can recommend publication in its current form.**

Thank you very much for your positive comments.